# Retinoic Acid-Induced Gene G(RIG-G) as a Novel Monitoring Biomarker in Leukemia and Its Clinical Applications

**DOI:** 10.3390/genes12071035

**Published:** 2021-07-02

**Authors:** Fei Wang, Jiale Tian, Li Pang, Junlu Wu, Anquan Shang, Zujun Sun, Dong Li, Jinsong Yan, Wenqiang Quan

**Affiliations:** 1Department of Neurosurgery, Shanghai Tongji Hospital, School of Medicine, Tongji University, Shanghai 200065, China; wangfeitj@tongji.edu.cn; 2Department of Laboratory Medicine, Shanghai Tongji Hospital, School of Medicine, Tongji University, Shanghai 200065, China; tianjiale@tongji.edu.cn (J.T.); pangli1980@126.com (L.P.); wujunlu@tongji.edu.cn (J.W.); shanganquan@tongji.edu.cn (A.S.); sunzujun@tongji.edu.cn (Z.S.); lidong@tongji.edu.cn (D.L.); 3Department of Hematology, Liaoning Medical Center for Hematopoietic Stem Cell Transplantation, Dalian Key Laboratory of Hematology, Liaoning Key Laboratory of Hematopoietic Stem Cell Transplantation and Translational Medicine, Dalian 116027, China; 4Diamond Bay Institute of Hematology, Second Hospital of Dalian Medical University, Dalian 116027, China

**Keywords:** real-time polymerase chain reaction, RIG-G gene, acute promyelocytic leukemia

## Abstract

Retinoic acid inducible gene G (*RIG-G*) is an inducible gene produced during the treatment of acute promyelocytic leukemia with all-trans retinoic acid (ATRA). However, it is unclear the expression level of RIG-G gene in the peripheral blood of healthy subjects and patients with acute promyelocytic leukemia (APL or AML-M3). In the present study, we established the TaqMan-MGB fluorescent probe qPCR (real-time polymerase chain reaction) method for the first time to detect the expression of *RIG-G* gene in APL. Twenty APL patients were selected, and their *RIG-G* expression levels were quantified to assess the correlation between the expression of peripheral blood and bone marrow samples. *U* test was used to analyze the expression level of *RIG-G* in the peripheral blood of 40 normal specimens and 20 APL patients to observe the prognostic monitoring effect of *RIG-G* gene in the ATRA treatment process. ROC (receiver operating characteristic curve) was used to analyze and test the diagnostic efficiency of RIG-G gene for APL patients. There is a strong positive correlation between the expression of *RIG-G* in peripheral blood and bone marrow of APL patients. The expression level of *RIG-G* in peripheral blood of APL patients is significantly lower than that in healthy controls (*p* < 0.001). The changes in the expression level of *RIG-G* in peripheral blood changed indicates the remission and recurrence of APL patients after ATRA treatment, and the ROC curve shows that it has a better diagnostic power for APL. In summary, the TaqMan-MGB real-time PCR method we have established has successfully run. The detection of *RIG-G* gene expression in peripheral blood can effectively monitor the disease changes of APL patients and avoid harmful bone marrow puncture injury.

## 1. Introduction

Acute promyelocytic leukemia (APL) is a distinct type of acute myeloid leukemia (AML), also known as sub-type AML-M3, with a high chance of developing severe complications, such as bleeding tendency, disseminated intravascular coagulation (DIC) and primary fibrinolytic hyperfunction, hence the mortality of patients with APL was extremely high; that is, until the mid-1980s when all-trans retinoic acid (ATRA) was adopted as a treatment, garnering much success [1]. At present, APL is characterized by specific t (15:17) chromosome translocation and PML-RARα gene fusion, which can be sufficiently rectified by the combined use of ATRA, anthracyclines, and arsenic trioxide (As_2_O_3_) [2,3,4]. Since the subsequent complete remission (CR), even free survival (EFS), and recovery rates for APL are relatively high, ATRA appears to be of high value for treatment of AML [5,6]. Nevertheless, the clinical application of ATRA has yet to achieve the ideal success rate, due to the emergence of impediments, such as the retinoic acid syndrome (RAS), retinoic acid resistance, and easy recurrence, etc. [7]. Moreover, its effect monitoring of patients has not been strictly evaluated. Although a laboratory assessment offers some guidance as to an APL diagnosis and prognosis through white blood cell count, bone marrow cell morphology, and t (15:17) chromosomal translocation and *PML-RAR**α* fusion gene assays, early and direct methods to indicate the treatment effects of ATRA on APL remain lacking, especially molecular biological methods that are effective, quick and simple. 

Retinoic acid-induced gene G (*RIG-G*) is an ATRA-inducible gene, which was originally isolated from the APL cell line NB4 by the technique of differential Display PCR [8]. Studies have shown that *RIG-G* is a growth inhibitor for APL cells, acting to upregulate the expression of cell cycle inhibitors p21 and p27, thereby contributing toward anti-proliferation and cell differentiation [9,10]. In untreated NB4 cells, the *RIG-G* mRNA levels are almost undetectable, but can be significantly enhanced by ATRA following 48–72 h of treatment. However, in NB4R1 cells, which are a part of a retinoic acid-resistant APL cell line, ATRA fails to induce the expression of *RIG-G* mRNA, and cell proliferation continues unaffected [8]. These results suggest that the expression level of *RIG-G* mRNA in APL cells serves as a suitable index the therapeutic effect of ATRA.

In this study, we devised a detection method for *RIG-G* mRNA expression and evaluated its clinical application to APL. We showed that *RIG-G* mRNA is a robust biomarker for diagnosis, prognosis judgment and relapse monitoring of patients with APL. 

## 2. Materials and Methods

### 2.1. Patients and Specimens

Twenty patients with APL admitted into the Department of Hematology, Shanghai Tongji Hospital, Tongji University School of Medicine over the period of January to December 2016 (all diagnosed using both bone marrow and peripheral blood samples at the same time) were selected for analysis. Peripheral blood samples obtained from 40 healthy subjects were relegated as the control, originating from 21 males and 19 females, with an average age of 41 ± 8 years old (from 15 to 81 years). These patients were ascertained to be free of major diseases, such as cancer and blood diseases. The recombinant plasmid pTRE-RIG-G, and NB4 and NB4R1 cell lines utilized were donated by Dr. Jianhua Tong. The study was approved by the ethics committee of Shanghai Tongji Hospital, Tongji University School of Medicine, and all the subjects granted informed consent (TONG Lun Sheng-KYSB-2014-02).

### 2.2. Total RNA Extraction, Quantification of Total RNA and cDNA Synthesis

The total RNAs were extracted using TRIZOL reagent (Invitrogen, San Francisco, CA, USA) in accordance with the manufacturer’s instructions. Briefly, mononuclear cells were first isolated from 2-mL samples of peripheral blood or bone marrow using Ficoll lymphocyte separation medium (density:1.077; Solarboi, Beijing, China) for centrifugation before being transferred into 1 mL of Trizol. Extracted total RNA was eluted in a final 50 µL of RNase-free water and stored at −80 °C. Quantification of total RNA was performed by measurement of A260 using a nucleic acid analyzer (DU730, Beckman-Coulter, CA, USA). The RNA samples reserved for downstream experiments were diluted to a uniform concentration. cDNA was then synthesized based on these RNA templates by reverse transcription (RT) using the PrimeScript^TM^ cDNA Synthesis Kit (Lot: RR047A, contains gDNA eraser, TaKaRa, Dalian, China). By using this kit, genomic DNA can be removed in 2 min at 42 °C. Then the RT mixture was composed of 5.0 µL of RNA template, 4.0 µL of 5 × RT buffer, 1.0 µL of PrimeScriptIIRTase (200 U/µL), 4.0 µL of dNTP mixture (each 2.5 mM), 0.5 µL of RNase inhibitor (40 U/µL), 1.0 µL of random 6 primer (50 µM), and 4.5 µL of sterilized DEPC water amounting to a final volume of 20 µL. Subsequently, the RT mixture was incubated at 65 °C for 5 min, 50 °C for 50 min, and 70 °C for 15 min in an ABI 7300 real-time PCR system machine (Applied Biosystems, Singapore).

### 2.3. Primer and TaqMan Probe Design

The complete nucleotide sequences of human *RIG-G* gene and human GAPDH were retrieved from the GenBank (https://www.ncbi.nlm.nih.gov/genbank/, accessed on 28 June 2021) database, and compared with DNASTAR software (DNASTAR, Madison, WI, USA). The primers and TaqMan probe were selected to locate in the highly conserved site of the non-coding region of human *RIG-G* gene. All primers used in the amplification are designed by Primer Premier 5.0 software (Applied Biosystems, San Francisco, CA, USA). The primers and TaqMan probe were synthesized and purified by Novo Biotechnology Co., Ltd (Shanghai, China). The 5′end of the probe is labeled with a reporter dye, and the 3′end is labeled with Quencher Dye (MGB). The sequences of primers and TaqMan probe used in this study are listed in Table 1. The PCR products were sequenced on the amplified fragments of *RIG-G* gene to confirm the specificity of amplification. Used the BLAST function of the NCBI public database to conduct a preliminary test of primer efficacy against the target fragment of *RIG-G* to be amplified (the length is 286 bp).

### 2.4. Standard Curve Determination

The recombinant plasmid pTRE-RIG-G was serially diluted in 10-fold increments in EASY dilution buffer (Takara, Dalian, China). Standard curves were generated by plotting the log of the starting quantity of template against the Ct (cycle threshold) values. Each dilution was tested in duplicate. The standard curves of X and Y axes were derived from the logarithm of concentration, and the corresponding slope of the curve as well as the correlation coefficient were obtained.

### 2.5. Real-time PCR for RIG-G Gene mRNA

Real-time PCR was performed using a Permix Ex Taq^TM^ Kit (Takara, China) on an ABI 7300 real-time PCR system machine (Applied Biosystems, San Francisco, CA, USA). The PCR reaction mixture consisted of 2 µL cDNA template, 10 µL Premix Ex Taq (Probe qPCR) (2×), 0.4 µL PCR forward primer (10 µM), 0.4 µL PCR reverse primer (10 µM), 0.8 µL of TaqMan probe (25 µM), 0.4 µL ROX Reference Dye (50×), and 6 µL of distilled water combined to a total volume of 20 µL. Real-time quantitative PCR (RT-PCR) was conducted in an ABI Prism 7300 as 40 cycles of pre-denaturation heating at 95 °C for 30 s, denaturation at 95 °C for 5 s, and annealing at 60 °C for 31 s. For quantification, a series of tenfold dilutions from 1 × 10^7^ to 1 × 10^1^ copies/µL of pTRE-RIG-G applicable standards (prepared as described above) and a negative control (distilled water) was run alongside the test samples.

### 2.6. Agarose Gel Electrophoresis

Prepared 2% agarose gel with 6 μL goldenview (Beyotime Biotechnology, Shanghai, China), shaken well. Mixed the agarose gel solution and carefully poured it onto the glass plate in the inner tan. Slowly spread the glue solution until a uniform glue layer was formed on the surface of the glass plate. Put the gel into the electrophoresis tank and add 1 × TAE agarose electrophoresis buffer until 1~2 mm below the gel plate. Added 20 µL qPCR product to 2 µL 10×loading, After sample loading, electrophoresis was performed at a voltage of 100 V for 30 min, then took pictures by gel imaging system (Bio-Rad, Hercules, CA, USA).

### 2.7. Morphological Staining of Bone Marrow Cells (Wright’s Staining)

Wright ’s staining (BASO, Zhuhai, China) was performed by following the steps recommended by the kit. In brief, first Wright’s Giemsa A solution (about 0.5 mL–0.8 mL) was dripped on the smear, and let the staining solution covered the entire specimen for 1 min; then Wright’s Giemsa B solution added on top of the A solution. Then it was blown out with ear wash ball to make the liquid surface rippling and make sure two liquids were fully mixed and dyed for 3–10 min. Rinsed with running water to prevent any sediment from depositing on the specimen; dried and checked with microscope.

### 2.8. PML-RARα Fusion Gene Detection

The quantitative real-time PCR detection of PML-RARα fusion gene was completed by Shanghai Xinchao Medical Laboratory. Briefly, collected 2 to 3 mL of EDTA anticoagulated bone marrow, and separated mononuclear cells by Ficoll density gradient centrifugation. Total RNA was extracted by Trizol method, and the concentration and purity of RNA were detected by UV spectrophotometer. Use RNA as template to detect the expression of PML/RARα fusion gene, ABL as a control gene. The primer design, reagents, etc. are provided by Shanghai Xinchao Medical Laboratory, and then proceed according to the operating instructions of the PML-RARα detection kit.

### 2.9. Statistical Analysis

The data were analyzed by SPSS25.0 software packages (SPSS Inc., Chicago, USA). If the continuous variable satisfies the normal distribution, it is represented by $\bar{x}$ ± s, and the discontinuous variable is represented by the median (interquartile range). The *χ^2^* test was used to compare the count data between groups. The Kolmogorov–Smirnov test showed that the expression level of *RIG-G* gene was a non-normal distribution of measurement data. Therefore, the Mann–Whitney *U* test was used for comparison between the two groups, and the GraphPad Prism 8 was used to draw the scatter plot. Using receiver operating characteristic (receiver operating characteristic, ROC) curve to evaluate the diagnostic efficacy of APL, *p* < 0.05 indicates a significant difference.

## 3. Results

### 3.1. Development of TaqMan-MGB Real-Time Quantitative PCR Assay for RIG-G Gene

The primers and TaqMan probe concentrations as well as reaction temperatures were optimized to establish an efficient real-time quantitative PCR assay for *RIG-G* gene expression levels. The TaqMan quantitative assay was carried out on a series of ten-fold dilutions from 1 × 10^7^ to 1 × 10^1^ copies/µL of pTRE-RIG-G applicable standard (prepared as described above) in duplicate, as the precursor to experimental sample analysis. The standard curve (Figure 1A) and the dynamic amplification curve (Figure 1B) were generated automatically. The standard curve showed the linear relationship between the log of the concentration of target RNA and the cycle threshold (Ct) value. The correlation coefficient of the standard curve was 0.999, indicating a near-exact log-linear relationship between the concentration of target RNA and the Ct value. The regression equation was y = −3.539x + 42.952, and the PCR amplification efficiency was 95.2%.

### 3.2. Methodological Performance Verification of the Established Method for RIG-G Gene Detection

Specificity is the capacity of the assay to detect specific target molecules [11]. The qPCR assay we arranged was run on the cDNA yielded the RNA extracts of peripheral blood samples from both normal subjects and APL patients, along with the plasmid standards and distilled water negative control. Melting curve analysis and 2% agarose gel electrophoresis provided verification of assay specificity, as was evident in the single peak present, representing the correct melting point of the target gene *RIG-G* and indicating that the PCR product was highly specific (Figure 1C). Following 2% agarose gel electrophoresis, all samples of plasmids (Lane 1~3), normal humans (Lane 4~6) and M3 patients (Lane 7~9) showed a band at 286 bp, while distilled water (Lane 10) showed no band as expected (Figure 1D). These results suggested that this method was highly specific.

Accuracy is a measure of the degree of closeness between the measured value and the true value [12]. In this study, we first sequenced the PCR products (APL patients) using the sanger sequencing method and found that the sequence was exactly the same as the *RIG-G* gene standard sequence (Figure 1E). Next, we used recombinant pTRE-RIG-G plasmid as the reference standard to determine the accuracy of the reference material. The concentration of 8 dilution standards (10^1^~10^8^ copies/μL) was measured by TaqMan probe method PCR, and the results were collated with the known values, giving a correlation coefficient of *r* = 0.999 and a bias of less than 5%.

Reproducibility is a measure of the consistency of a test. The pTRE-RIG-G plasmid was diluted into 3 concentrations (10^7^ copies/μL, 10^4^ copies/μL, and 10^1^ copies/μL) in preparation for the following assessments: Intra-batch precision: 3 concentrations of the samples were repeated 20 times; inter-run precision: continuous determination for 5 days (different staff, different batches of primers), repeated 4 times every day. Results showed that the intra-batch precision for the three concentrations gave CV values of 1.38%, 2.31% and 7.56%, respectively, and the inter-run precision yielded CV values was 0.71%, 1.17% and 5.07%, respectively, indicating that our method supplied a reliable reproducibility.

Analytical sensitivity is represented by the minimum analyte concentration detectable in a system or by a method, also known as the lowest detection limit (LOD) [11]. Plasmids were diluted to 10^3^, 10^2^, 10^1^, and 10^0^ copies/μL for examination of each concentration 20 times, and the result coincidence rate was found to be 75% at 10^0^ copies/μL and greater than 95% at 10^1^ copies/μL, suggesting that the detection sensitivity of the method extended to 10^1^ copies/μL. 

### 3.3. Influence of Interfering Substances on the Detection of RIG-G Gene

In consideration of interfering substances on *RIG-G* mRNA levels, 12 samples of hyperbilirubinemia specimens (total bilirubin range: 168~569 mol/L) and 12 cases of hyperlipidemia (triglyceride range 3.99~21.78 mmol/L) were inspected through the established TaqMan-MGB real-time fluorescence quantitative PCR method. Total RNAs were isolated by direct RNA extraction, or RNA extraction after removal of bilirubin and blood lipids via plasma exchange. The RNA concentration of hyperbilirubinemia specimens treated with plasma exchange was (293.5 ± 120.6) ng/uL, and the RNA concentration of original specimens (300.9 ± 128.1) ng/ul, the difference between RNA concentrations has not statistically significant (*t* = 0.145, *p* = 0.886; Figure 2A); the difference between the results of detecting *RIG-G* gene expression between the two was also not statistically significant (*U* = 72, *p* = 0.989; Figure 2C). The RNA concentration of the original lipemia specimen (when triglycerides < 21.78 mmol/L) was (288.1 ± 105.3) ng/uL, the RNA concentration after replacement was (294.2 ± 109.6) ng/ul, the difference between RNA concentration There was no statistical significance (*t* = 0.140, *p* = 0.890; Figure 2B); the difference between the results of multiple detections of *RIG-G* gene expression was also not statistically significant (*U* = 71.5, *p* > 0.999; Figure 2D);As expected, neither hyperbilirubinemia nor hyperlipidemia exerted a significant effect on the concentration of RNA or the expression of *RIG-G* mRNA.

### 3.4. Evaluation of Established PCR Method for Detection of RIG-G mRNA in Peripheral Blood of APL Patients

In order to clarify the correlation between the detection of *RIG-G* gene in peripheral blood and bone marrow, we also selected bone marrow and peripheral blood samples from 12 cases and used the established method for verification. We compared the levels of *RIG-G* mRNA in peripheral blood against those in bone marrow samples and found that the correlation coefficient between the two was *r* = 0.992, the slope was *b* = 0.973, and a paired *t* test with *p* > 0.05. It is thus evident that there is a close relationship between the expression of *RIG-G* mRNA in peripheral blood and that in bone marrow in patients with APL (Figure 3A). Next, A total of 60 cases (20 patients with APL confirmed by bone marrow cell morphology method and 40 healthy donors) were analyzed by the established PCR for *RIG-G* mRNA in peripheral blood. The results showed that APL patients {1.987 × 10^3^ (0.723 × 10^3^~5.594 × 10^3^) copies/μL} exhibited a lower expression of *RIG-G* mRNA than healthy donors{2.748 × 10^4^ (1.585 × 10^4^~4.71 × 10^4^) copies /μL} (*p* < 0.001, Figure 3B).

Bone marrow cell morphology is the gold standard for the diagnosis of APL patients. The consistency of diagnosis between *RIG-G* mRNA and bone marrow cell morphology as biomarkers of APL was evaluated by a kappa (*χ^2^*) test (Table 2). The kappa value constitutes the internal consistency coefficient, which was the main index used to determine the consistency of the two methods. A kappa value of 0.883 was obtained, indicating that the consistency of diagnosis based on *RIG-G* mRNA versus bone marrow cell morphology for APL was of a high degree (Table 2).

### 3.5. Relationship between RIG-G mRNA Expression and Prognosis in APL Patients

A 26-month follow-up study on the dynamics of *RIG-G* mRNA in peripheral blood was conducted on APL patients undergoing primary and all-trans retinoic acid treatment. As can be seen, the expression level of *RIG-G* mRNA was extremely low at the onset of APL, but significantly increased after ATRA-treatment (*n* = 7; *p*
*=* 0.001), and it was lower again in 3 recurrence patients (*p* = 0.017; Figure 4A). We observed 3 APL cases, the electrophoresis results showed patient 1 (Lane 1 in Figure 4B), with bone marrow promyelocytic cell numbers up at 91.5%. All-trans retinoic acid treatment lasting 10 months aided the patient into remission, which was apparent in the rise in *RIG-G* gene level expression (Lane 2 in Figure 4B), as well as the reduction in the number of promyelocytic cells in bone marrow to 3.5% (Figure 4C). Patient 2 was similar to patient 1, with *RIG-G* expression extremely low in the beginning (Lane 4 in Figure 4B). After 15 months of all-trans retinoic acid treatment, *RIG-G* expression increased (Lane5 in Figure 4B), and bone marrow promyelocytic cells decreased. In another case, the expression of *RIG-G* mRNA level was again extremely low at the onset of APL (Lane 7 in Figure 4B), with bone marrow promyelocytic cells reaching 81%. Following 7 months of treatment with all-trans retinoic acid, the patient entered remission, with *RIG-G* gene levels having increased accordingly (Lane 8 in Figure 4B). However, 26 months later, relapse occurred, causing the promyelocytic cell numbers to increase to 15% in the bone marrow and the expression level of *RIG-G* mRNA to become low again (Lane 9 in Figure 4B).

A receiver operating characteristic curve (ROC) analysis of *RIG-G* mRNA in peripheral blood of APL patients indicated that the cutoff value was 5.50 × 10^3^ copies/μL, and the area under the curve (*AUC*) of *RIG-G* was 0.974, signifying *RIG-G* mRNA as a strong predictor of APL, with 83.33% sensitivity (17/20) and 100% specificity (20/20) (Figure 4D). 

### 3.6. Correlation of RIG-G Gene Expression with PML-RARα Fusion Gene and Blast Cells in APL

We performed a statistical analysis on the correlation between the *RIG-G* gene, PML-RARα/ABL (%), and Blast (%) of 11 APL patients before and after ATRA treatment. The results showed that *RIG-G* gene expression and PML-RARα/ABL (%) were highly negatively correlated before ATRA treatment (r = −0.909, *p* = 0.0003, Figure 5A), *RIG-G* gene expression and Blast (%) in peripheral blood were slightly negatively correlated before ATRA treatment (r = −0.300, *p* = 0.371 Figure 5B). Due to the PML-RARα/ABL (%) and Blast (%) values in most APL patients were reduced to 0% after ATRA treatment, we cannot accurately analyze the correlation between the two results after remission and *RIG-G* gene expression (results not shown). Therefore, we statistically analyzed the changes of *RIG-G* gene, PML-RARα/ABL (%), Blast (%) before and after ATRA treatment in each APL patient (Figure 5C–E). The results showed that PML-RARα/ABL (%), Blast (%) decreased significantly after ATRA treatment (*p* < 0.0001). At the same time, the expression level of *RIG-G* gene in the peripheral blood of APL increased significantly (*p* = 0.0004). This indirectly showed that the expression of *RIG-G* gene changed with the changes of PML-RARα gene and Blast cells.

### 3.7. RIG-G Gene Expression Is Reduced in Patients with Many Types of Leukemia

The *RIG-G* gene is found in B acute lymphocytic leukemia (B-ALL), T acute lymphocytic leukemia (T-ALL), Chronic lymphocytic leukemia (CLL), Acute myeloid leukemia M0 (AML-M0), Acute myelogenous leukemia M2 (AML-M2), Acute myelogenous leukemia M4 (AML-M4), Chronic myelogenous leukemia (CML), Myelodysplastic Syndrome (MDS), Non-Hodgkin Lymphoma (NHL), etc. The peripheral blood of patients with untreated other types of hematological tumors all have varying degrees of low expression (Figure 6A,B), which is also consistent with our previous basic research results, confirming that the RIG-G gene is an important tumor suppressor gene [13].

## 4. Discussion

In the mid-1980s, the application of ATRA to induce differentiation of cells in patients with AML-M3 leukemia proved to be an effective treatment [1]. However, very few methods are available to monitor the advancement of a prognosis and the ensuing efficacy of retinoic acid treatment. Although white blood cell count and the percentage of abnormal promyelocytic cells in bone marrow are important prognostic factors in clinical practice, the reliability of the criterion is insufficient, and additionally influenced by subjective judgment, dependent on the experience of the individual examiner [14]. Moreover, it is difficult to standardize appraisals of an APL prognosis, which also fail to expediently register the effect of ATRA treatment on APL patients. Therefore, a fast, effective, simple and accurate method must be procured to evaluate the prognosis of APL patients, which will assist in directing to the clinical treatment of APL as appropriate. 

It has been suggested that *RIG-G* inhibits the proliferation of tumor cells and promotes the differentiation of promyelocytic cells [15,16,17]. Our study found that the expression level of *RIG-G* mRNA was closely related to the disease status of patients with APL. The area under the ROC curve analysis for *RIG-G* as determined by TaqMan-MGB real-time quantitative PCR was 0.974 for APL, denoting the usefulness of *RIG-G* in the detection of APL. The method of TaqMan-MGB real-time quantitative PCR we established to detect the level of *RIG-G* mRNA displayed a high sensitivity, with a minimum detection limit of 10^1^ copies/μL, excellent stability and repeatability, as well as a wide linear detection range of up to 7 orders of magnitude (10^7^~10^1^ copies/μL) and was also not affected by high bilirubin and hyperlipidemia. This proposed method not only utilizes the beneficial characteristics of the traditional TaqMan probe, but also possesses some new advantages, such as improving the signal to noise ratio (SNR), eliminating background fluorescence, and demonstrating higher quenching efficiency. 

In carrying out the described qPCR method, there are several concerns of which to be wary. Chloroform, phenol, and latex gloves with talc act as inhibitors of the PCR amplification reaction, specifically blocking Taq DNA polymerase activity or polymerase binding to target DNA, often leading to false negative results [18,19,20,21]. Heparin anticoagulant whole blood specimens are not eligible for the PCR amplification reaction due to having an inhibitory effect on the Taq DNA polymerase [22]. On the other hand, EDTA-K2 anticoagulant is suitable for use as it does not affect cell count and size, nor the morphology of red blood cells while preventing platelet aggregation as desired. More notably, it does not contain any substances with inhibitory activity against the Taq enzyme that would affect the PCR reaction [23]. Another consideration to take into account is the necessary strict observance of operation and cleaning protocols to avoid cross contamination of samples in the process of nucleic acid extraction, which prevents the outcome of false positive results [24].

Our study also found that the expression level of *RIG-G* mRNA was associated with disease stage in APL patients. Namely, *RIG-G* mRNA expression was lower in patients who had not been treated or had relapsed, but higher in patients in complete remission. For untreated or relapsed APL patients, the number of primitive cells in bone marrow cells increased, whereas for patients in complete remission, cell numbers were significantly lower. At the same time, the expression level of *RIG-G* mRNA is extremely negatively correlated with the change of PML-RARα fusion gene. As the level of PML-RARα decreases, the level of *RIG-G* mRNA increases after ATRA treatment. Thus, *RIG-G* mRNA levels are higher in mature cells and lower in primary cells. Our previous studies found that the cytoplasm of normal granulocytes is replete with *RIG-G* protein, suggesting that RIG-G may play a role in the differentiation of bone marrow cells. There is the possibility that low levels of *RIG-G* gene are due to abnormal conditions in APL cells, such as the negative regulation by PML-RAR of RIG-G expression.

Our study also revealed that in addition to the expression of *RIG-G* gene in peripheral blood having a strong correlation with that in bone marrow in patients with APL, the expression level of *RIG-G* mRNA in peripheral blood was highly consistent with bone marrow cytology as well. Therefore, the detection of *RIG-G* mRNA can be used to evaluate the M3 prognosis accuracy and efficacy of treatment with retinoic acid. Furthermore, *RIG-G* gene can be directly monitored through peripheral blood, which excludes the need to perform pain-inducing bone marrow punctures on patients, thereby facilitating regular and long-term monitoring.

In addition, it is worth noting that there are multiple transcription variants of RIG-G gene (IFIT3-201, IFIT3-202, IFIT-206). When studying the function of different transcription variants, we should find a specific transcription variant sequence relative to the other variants, then design primers for this sequence. In this case, other transcription variants will not be amplified by this pair of primers. In our study, the primers we designed are for the conservative sequence of the *RIG-G* gene, that is, the consensus sequence of the three transcripts, so it is impossible to distinguish the individual transcript variants of the *RIG-G* gene. However, as this study is a clinical prospective study, qRT-PCR detects the overall expression level of the gene and does not involve the function of each transcript of the *RIG-G* gene. Therefore, by detecting changes in the expression level of the conservative sequence of the *RIG-G* gene, we can evaluate the patient’s condition and provide diagnostic advice for clinical diagnosis and treatment. If we are going to study the function of the *RIG-G* gene, the influence of different transcription variants on gene function should be considered.

In conclusion, the TaqMan-MGB real-time fluorescence quantitative PCR assay established in this study can be used to detect *RIG-G* mRNA in peripheral blood rapidly and accurately. Our method has the advantages of high sensitivity, strong specificity, good repeatability, wide linear range and fewer interference factors. The expression level of *RIG-G* mRNA is associated with the disease status of APL patients, and can thus serve as a potential biomarker for diagnosis, prognostic judgment and recurrence monitoring of APL.

## Figures and Tables

**Figure 1 genes-12-01035-f001:**
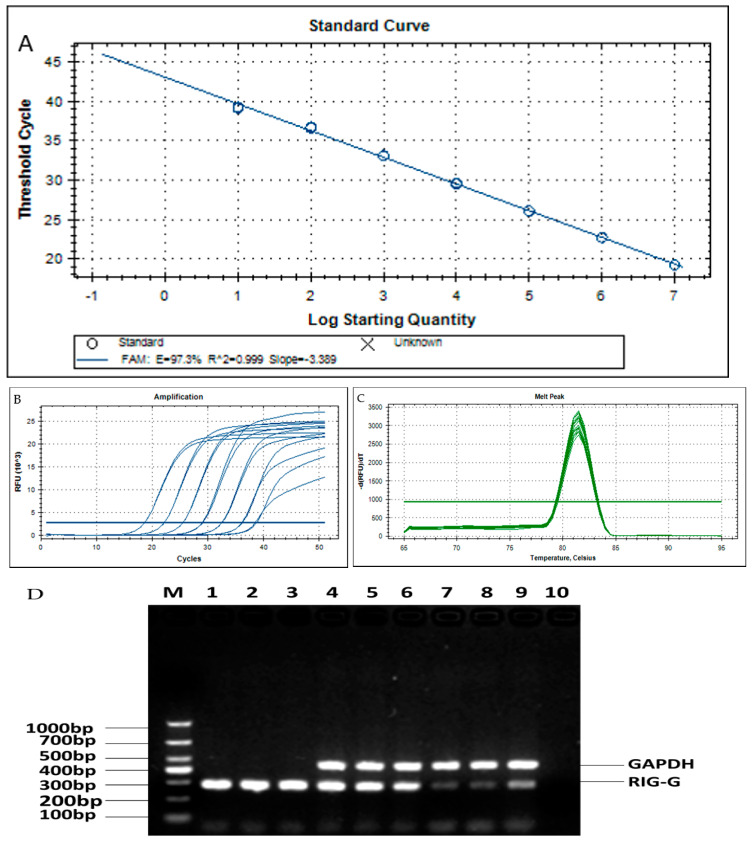
Successfully established TaqMan-MGB *RIG-G* gene mRNA real-time quantitative PCR detection method. (**A**) The linear relationship between log of concentration and Ct values for a series of dilutions from 1 × 10^7^ to 1 × 10^1^ of RIG-G amplicon standard (R^2^ = 0.999). Each dilution of RIG-G plasmid amplicon standard was assayed in duplicate by TaqMan-MGB real-time fluorescence quantitative PCR. (**B**) Serial dilutions from 1 × 10^7^ to 1 × 10^1^ of RIG-G plasmid amplicon standards, the curves from left to right correspond to 1 × 10^7^, 1 × 10^6^, 1 × 10^5^, 1 × 10^4^, 1 × 10^3^, 1 × 10^2^, 1 × 10^1^ copies/uL, respectively. (**C**) RIG-G gene mRNA melting curve. (**D**) Detection of PCR amplification products by 2% agarose gel electrophoresis. M: marker; 1~3: plasmid; 4~6: normal; 7~9: M3 patients; 10: sterilized double distilled water. (**E**)The PCR products were sequenced using the Sanger method, and the results showed that the sequence of RIG-G gene was completely consistent.

**Figure 2 genes-12-01035-f002:**
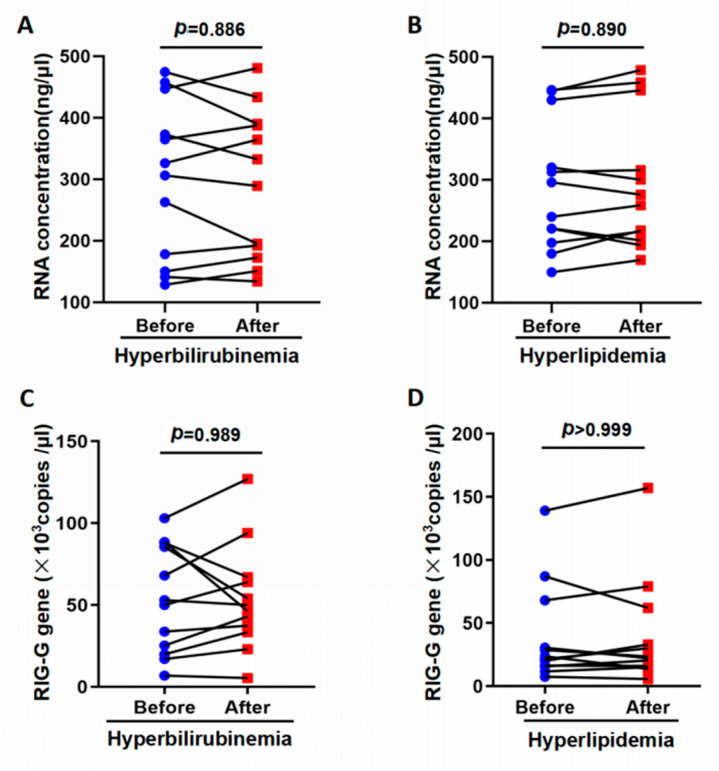
Hyperbilirubinemia and hyperlipidemia have no significant interference in detecting RIG gene in peripheral blood. (**A**,**B**) Hyperbilirubinemia and hyperlipidemia samples were obtained by direct extraction of RNA or extraction following plasma exchange to remove bilirubin and lipids. The results are expressed as the mean ± SD (*n* = 12, *p* > 0.05). (**C**,**D**) Use of the established qPCR method to detect RIG-G gene, showed that hyperbilirubinemia and hyperlipidemia had no significant effect (n = 12, *p >* 0.05).

**Figure 3 genes-12-01035-f003:**
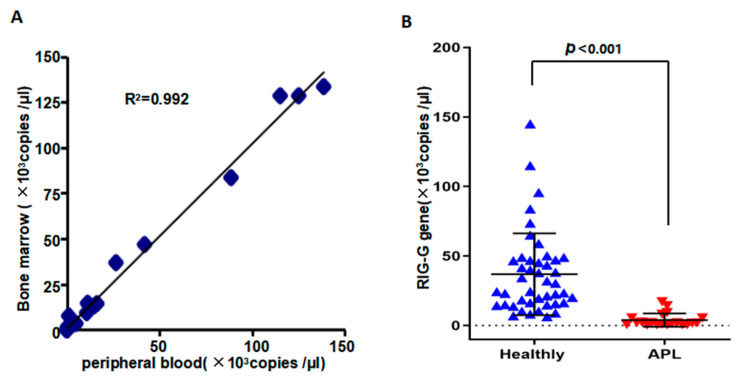
The expression and correlation of RIG-G in healthy donors and APL (M3) patients. (**A**) The correlation of RIG-G gene expression in peripheral blood and bone marrow showed no significant difference (*n* = 12, *p*
*>* 0.05), indicated that consistency of two methods was good (R^2^ = 0.992). (**B**) The expression of RIG-G gene in peripheral blood of APL patients was markedly lower than that of healthy control group, with a difference that was statistically significant (*p* < 0.001).

**Figure 4 genes-12-01035-f004:**
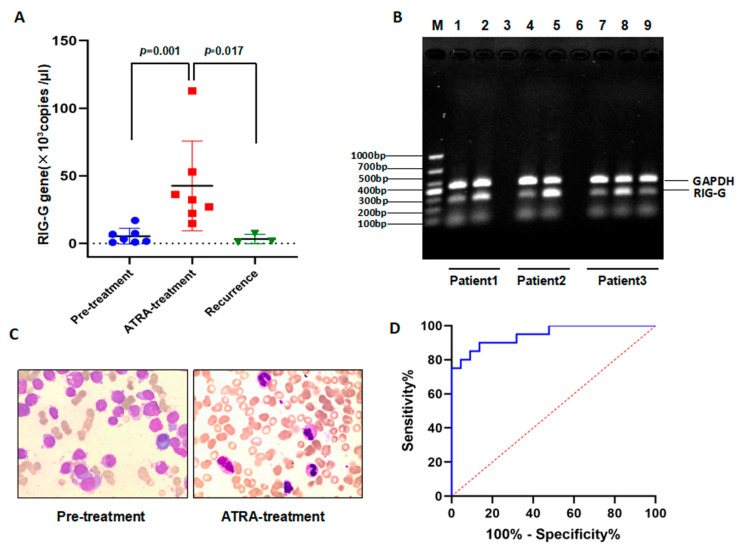
The prognosis monitoring application and diagnostic value of *RIG-G gene* in APL patients. (**A**)After ATRA-treatment, the expression of RIG-G gene in peripheral blood of APL patients was significantly increased (*n* = 7, *p* = 0.001), But it was lower again in 3 recurrence patients (*p* = 0.017). (**B**) Detection PCR amplification of RIG-G gene products by 2% agarose gel electrophoresis of 3 APL patients (lane 1,4,7 = Pre-treatment; lane 2,5,8 = ATRA-treatment; lane 9 = Recurrence). (**C**) Changes of promyelocytic cells in bone marrow of patients with APL. (**D**)ROC curve to the diagnostic efficacy of RIG-G gene in APL patients.

**Figure 5 genes-12-01035-f005:**
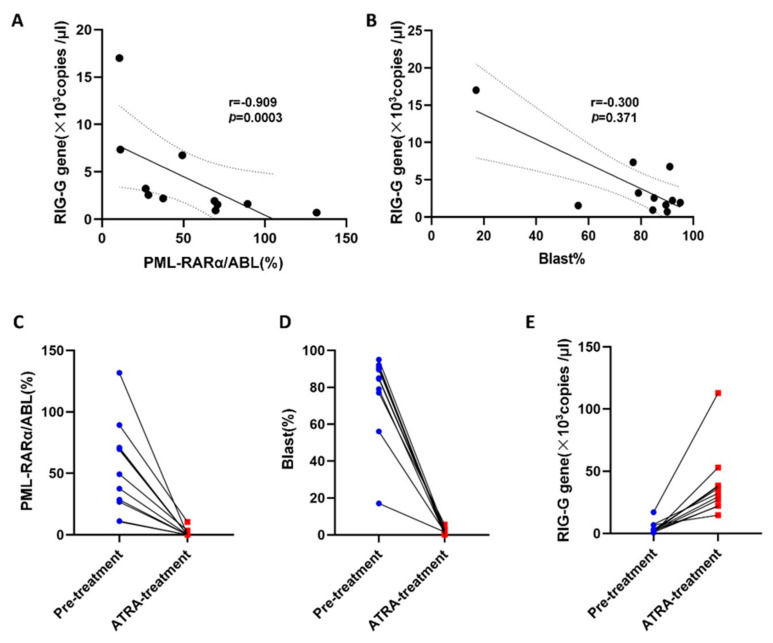
Correlation of RIG-G gene expression with PML-RARα fusion gene and blast cells in APL. (**A**) RIG-G gene expression and PML-RARα/ABL (%) were highly negatively correlated before ATRA treatment (*n* = 11, r = −0.909, *p* = 0.0003).(**B**) RIG-G gene expression and Blast (%) were slightly negatively correlated before ATRA treatment (*n* = 11, r = −0.300, *p* = 0.371). (**C**–**E**) PML-RARα/ABL (%) and Blast (%) decreased significantly after ATRA treatment (*p* < 0.0001), meanwhile, the expression level of RIG-G gene increased significantly (*p* = 0.0004) in APL patients.

**Figure 6 genes-12-01035-f006:**
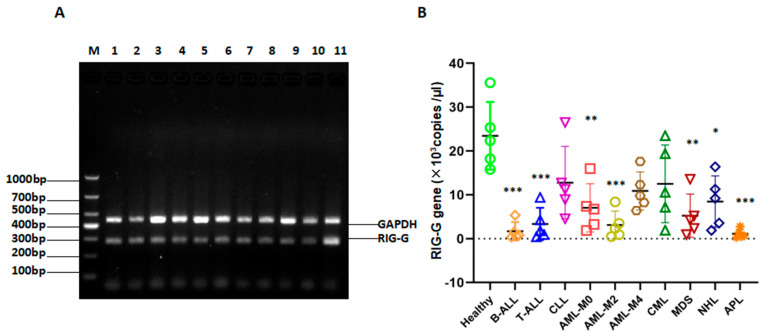
Relationship between RIG-G gene expression in multiple types of leukemia patients. (**A**) Detection PCR amplification products of RIG-G gene by 2% agarose gel electrophoresis of multiple types of leukemia patients (lane 1 = B-ALL; lane 2 = T-ALL; lane 3 = CLL; lane 4 = AML-M0; lane 5 = AML-M2; lane 6 = AML-M4; lane 7 = CML; lane 8 = MDS; lane 9 = NHL; lane10 = APL; lane11 = healthy control). (**B**) The RIG-G gene expression levels in multiple types of leukemia patients (* *p*< 0.05, ** *p* < 0.01, *** *p* < 0.001).

**Table 1 genes-12-01035-t001:** List of primers and TaqMan-MGB probe used for reverse transcriptionquantitative polymerase chain reaction analysis.

Gene/Probe	Primer Sequence
RIG-G	F: 5′-GAAGAAATGAAAGGGCGAAGG-3′
	R: 5′-AGGACATCTGTTTGGCAAGGAG-3′
GAPDH	F: 5′- AGGTCGGAGTCAACGGATTTGGT-3′
	R: 5′- GTGCAGGAGGCATTGCTGATGAT-3′
Taqman RIG-G-probe	5′-FAM-AGGACTCAGCTCAATGG-MGB-3′

Abbreviation: F, forward; R, Reverse.

**Table 2 genes-12-01035-t002:** Consistency of RIG-G gene expression and bone marrow cell morphology.

Assay	Gold Standard			Sum Total
	APL(+)		APL(-)	
**RIG-G gene**	High	17	0	17
	Low	3	40	43
**Sum total**		20	40	60

## Data Availability

The data that support the findings of this study are available from the corresponding author upon reasonable request.

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
