# Peer review of "Retinoic Acid-Induced Gene G(RIG-G) as a Novel Monitoring Biomarker in Leukemia and Its Clinical Applications"

_genes, 2021, doi:10.3390/genes12071035_

Round 1
Reviewer 1 Report
Thank you for your reply and the revisions.
Author Response
Thanks for your comments and suggestions.
Reviewer 2 Report
The authors have adequately responded to my concerns.
Author Response

(The authors gave the same response as above.)

Reviewer 3 Report
A major concern in the design of the RT-PCR Taqman assay is the forward and reverse primers are located within the same exon of IFIT3 (RIG-G). This design does not exclude genomic DNA from being amplified along with the cDNA products. If the primers were separated by a large intron (ie., 10.5kb) as found in IFIT3-202 and IFIT3-206, which appear to be to most abundant isomers of the IFIT3 transcript variants, it is possible that contaminant DNA would not be efficiently amplified (and therefore not a significant problem). Alternatively, contaminating genomic DNA can be digested by DNase from each of the samples, but this protocol requires verification that genomic DNA has been removed (no product present after PCR amplification without RT).
In addition to the design of the primers, some consideration of the expression levels of all the transcript variants (IFIT3-201, IFIT3-202, IFIT-206) needs to considered in response to the location of the primer assay. The authors should be especially wary of the exon structure of the IFIT3 gene in the design of any primers covering each of these variants.
Author Response
--Thanks to the reviewer’s critics. Yes, we have chosen a commercial kit during the reverse transcription(RT) phase of the experiment, which can ensure that the interference of gDNA is removed before RT, and finally, it can synthesize cDNA with low background, high purity and good integrity. Moreover, we also verified the specificity of the PCR product (as shown in the Figure.1 below), no product present after PCR amplification without RT (lane7, 8). As the reviewer mentioned, in the follow-up research, we will pay more attention to the design of primers to eliminate interference to experimental results.

Round 2
Reviewer 3 Report
In response to my query concerning the presence of contaminating DNA, you have stated that the extraction kit that you chose is a kit that excludes gDNA before RT. However, you list in Materials and Methods the Invitrogen kit, Trizol, as the extraction system. This kit does not remove contaminating DNA unless titrated (carefully determined amounts) of DNase is used as an extra step. In your figure supplied with your response, only two samples (lanes 7 and 8) are shown that represent PCR amplification without RT along with 6 samples with RT (Lanes 1-6). This is insufficient information. Only two samples of unknown identity (not matched) as your proof that the Trizol extraction kit (somehow) excluded genomic DNA. I'm not convinced by this explanation or the evidence from only two unmatched samples. A second point raised in the review is consideration of the expression levels of all the transcript variants (IFIT3-201, IFIT3-202, IFIT-206) in response to the location of the primer assay. You have not addressed this issue in the current manuscript. Instead, you have simply decided that primer design to consider variant contributions would be improved in future papers??Author Response
--Thanks to the reviewer’s critics. First,I may not make it clear in the method. In fact, removing gDNA is not at the stage of extracting RNA. We did it at the stage of reverse transcription(RT), we had chosen a commercial kit during the reverse transcription phase of the experiment, This kit provided by Takara company, Catalog: RR047A, contains gDNA eraser, which can remove gDNA in just 2 minutes at 42℃. We also add a description of this simple step in the method section.
The second point, we suggest that the primers designed in this study are conservative sequences of the RIG-G gene, that is, the common sequences of three transcripts (IFIT3-201, IFIT3-202, IFIT-206). Therefore, qRT-PCR detects the overall level of RIG-G gene expression, and does not involve the expression of a particular transcript. Moreover, as a prospective clinical study, this study did not involve the transcription functions of the gene. In order to provide diagnostic basis for clinical diagnosis and treatment, we evaluated the condition of patients by detecting the expression level of RIG-G gene conservative sequence. Of course, this is also a limitation of the primer design, the design of a primer for a particular transcriptional variant so that it can not specifically distinguish the changes and functions of a particular transcriptional variant. In future studies of the RIG-G function, it is important to pay attention to the design of primers in order to distinguish functional differences in transcriptional variants. To sum up, in the discussion section of this manuscript, we propose the limitations of this study and add a discussion on primer design for different transcription variants.

This manuscript is a resubmission of an earlier submission. The following is a list of the peer review reports and author responses from that submission.
Round 1
Reviewer 1 Report
In their work the authors present RIG-G as novel biomarker for monitoring APL by establishing a real-time PCR assay for measuring RIG-G in the peripheral blood.The method was established and tested thoroughly. Overall, some minor aspects need improvement:
- are baseline measurements at first diagnosis needed to assess the response? Since the gene is expressed in healthy persons one would need a patient based control.
- in chapter 3.4 you point to a prognostic relationship between RIG-G expression and APL. In case of relapse a lower espression of the gene is observed. This correlates with higher cell count of leukemic cells in the bone marrow. Although this also correlates with the relapse status verifying the lower expression levels during active disease, no prognostic relevance could be drawn from this result. It would be of interest, e.g. if the time to response or the amount of transcript reduction measured by gene expression is of prognostic relevance
- as displayed in Figure 5 the expression of RIG-G is not specific for APL, are the mRNA levels rising under treatment in other entities as well?
- please put the novel monitoring of RIG-G into the context of molecular, PCR based monitoring of PML/RARA. Since this can performed in peripheral blood by quite sensitive methods e.g. ddPCR it would be of interest, if measuring RIG-G expression is helpful in that context.
- overall, please check the grammar (abstract), the blanks and brackets
Author Response
The point-to-point reply to the reviewers’ questions and suggestions as below.
#reviewer 1
In their work the authors present RIG-G as novel biomarker for monitoring APL by establishing a real-time PCR assay for measuring RIG-G in the peripheral blood.The method was established and tested thoroughly. Overall, some minor aspects need improvement:
are baseline measurements at first diagnosis needed to assess the response? Since the gene is expressed in healthy persons one would need a patient based control.
--Thanks to the reviewer’s suggestion. In the results of Figure 3B of this article, we described the expression of RIG-G gene in healthy people and the expression of patients with first-onset APL.
in chapter 3.4 you point to a prognostic relationship between RIG-G expression and APL. In case of relapse a lower expression of the gene is observed. This correlates with higher cell count of leukemic cells in the bone marrow. Although this also correlates with the relapse status verifying the lower expression levels during active disease, no prognostic relevance could be drawn from this result. It would be of interest, e.g. if the time to response or the amount of transcript reduction measured by gene expression is of prognostic relevance as displayed in Figure 5 the expression of RIG-G is not specific for APL, are the mRNA levels rising under treatment in other entities as well?
--Thanks to the reviewer’s suggestion. The RIG-G gene has been confirmed to be low-expressed in some solid tumors (Li D, Sun J, Liu W, et al. Rig-G is a growth inhibitory factor of lung cancer cells that suppresses STAT3 and NF-κB. Oncotarget. 2016;7(40):66032-66050. doi:10.18632/oncotarget.11797),Our latest research found that it may also be an important tumor suppressor gene(Sun J, Wang X, Liu W, et al. Novel evidence for retinoic acid-induced G (Rig-G) as a tumor suppressor by activating p53 signaling pathway in lung cancer. FASEB J. 2020;34(9):11900-11912. doi:10.1096/fj.201903220R). In Figure 5, we used the Taqman method established in this article to detect the RIG-G gene expression of some collected patients with other types of hematological tumors, and some of them have a decreased RIG-G gene expression without treatment. However, because the number of patients with other hematological diseases selected in this study is very limited, We have not conducted long-term follow-up treatment for these cases, so we cannot infer the changes of the RIG-G gene in the peripheral blood.
please put the novel monitoring of RIG-G into the context of molecular, PCR based monitoring of PML/RARA. Since this can performed in peripheral blood by quite sensitive methods e.g. ddPCR it would be of interest, if measuring RIG-G expression is helpful in that context.
--Thanks to the reviewer’s suggestion. Recently, our collaboration research team has established a ddPCR method for monitoring PML/RARA genes (Zhu H, Wang ZY, Ding XQ, Wang RX, Pan XR, Tong JH. Zhongguo Shi Yan Xue Ye Xue Za Zhi. 2019;27( 3):747-752. doi:10.19746/j.cnki.issn.1009-2137.2019.03.017. We will also consider applying this sensitive technology to the detection of RIG-G gene in our follow-up research, and further discuss the consistency and relevance of the two methods.
overall, please check the grammar (abstract), the blanks and brackets
--Thanks to the reviewer’s suggestion.We have carefully checked the manuscript and have made corresponding amendments to the grammar,the blanks and brackets as required.
Reviewer 2 Report
In this manuscript, the authors tested whether expression of RIG-G expression by TaqMan real-time PCR can be used to assess therapy response in patients with APL. This is based on previous work indicating that RIG-G expression increased with ATRA treatment in NB4 cells, but did not change in an ATRA-resistant subclone from that cell line. The authors showed careful evaluation of their assay and results from a small cohort of 20 patients, including samples from before and after ATRA treatment. While as a group, RIG-G expression significantly increased with ATRA, they only show the change in RIG-G expression (although not quantitated) and how this change correlated with the percentage of blast cells for a 3 select patients. Therefore, it is difficult to predict whether RIG expression will be a valuable tool to assess therapy response for individual patients.
Major concerns:
- They do not show how change in RIG correlates with change in the percentage of blasts cells. In figure 4 A, there appears to be patients with low RIG-G expression after ATRA, as well as patients with very high RIG-G expression. Did these changes correlates with changes in % of blast cells? Without such data for the whole cohort, it is difficult to understand how RIG-G expression can be used to monitor therapy response.
- The labeling of Table 2 is unclear. It appears that patient samples are divided into APL(+) and APL(-) groups based on bone marrow histology, and also by RIG-G expression levels. If so, it would be more appropriate to label the grouping based on RIG-G expression as "high" and "low", as RIG-G expression is not an accepted method to diagnose APL, and the cutoff between the two groups should be explained. Also, it appears that 3 patients were diagnosed with APL by bone marrow histology, but showed RIG-G expression consistent with healthy patients. The authors should elaborate on this finding.
Minor Concern:
1. In Figure 1 E, it is unclear which PCR products were sequenced and how many samples were sequenced.
Author Response
In this manuscript, the authors tested whether expression of RIG-G expression by TaqMan real-time PCR can be used to assess therapy response in patients with APL. This is based on previous work indicating that RIG-G expression increased with ATRA treatment in NB4 cells, but did not change in an ATRA-resistant subclone from that cell line. The authors showed careful evaluation of their assay and results from a small cohort of 20 patients, including samples from before and after ATRA treatment. While as a group, RIG-G expression significantly increased with ATRA, they only show the change in RIG-G expression (although not quantitated) and how this change correlated with the percentage of blast cells for a 3 select patients. Therefore, it is difficult to predict whether RIG expression will be a valuable tool to assess therapy response for individual patients.
Major concerns:
They do not show how change in RIG correlates with change in the percentage of blasts cells. In figure 4 A, there appears to be patients with low RIG-G expression after ATRA, as well as patients with very high RIG-G expression. Did these changes correlates with changes in % of blast cells? Without such data for the whole cohort, it is difficult to understand how RIG-G expression can be used to monitor therapy response.
--Thanks to the reviewer’s suggestion. We have observed 7 APL patients continuously for a long time. Due to individual differences, they have different changes in RIG-G after ATRA treatment. But they are all higher than a certain value {4.259×104 (2.22×104~5.29×104) copies /μl} (Figure 4A). We observed that APL patients with low RIG-G expression seem to have more blast cells in the bone marrow. ATRA treatment may take longer, but this phenomenon is temporarily not statistically significant. The specific relationship between the expression of RIG-G gene and blast cells may require us to further collect APL patients for verification of large samples.
The labeling of Table 2 is unclear. It appears that patient samples are divided into APL(+) and APL(-) groups based on bone marrow histology, and also by RIG-G expression levels. If so, it would be more appropriate to label the grouping based on RIG-G expression as "high" and "low", as RIG-G expression is not an accepted method to diagnose APL, and the cutoff between the two groups should be explained. Also, it appears that 3 patients were diagnosed with APL by bone marrow histology, but showed RIG-G expression consistent with healthy patients. The authors should elaborate on this finding.
--Thanks to the reviewer’s suggestion. The groups of RIG-G gene expressions have been marked as "high" and "low" in Table 2. The higher expression of RIG-G in the peripheral blood of 3 patients with APL and the inconsistency with bone marrow results may be due to tumor heterogeneity, that is, the differences between individual tumor patients. At present, the majority of tumor markers cannot achieve detection 100% specificity or 100% sensitivity.
Minor Concern:
- In Figure1 E, it is unclear which PCR products were sequenced and how many samples were sequenced.
--Thanks to the reviewer’ critics. The sequenced samples were the plasmids (Lane 1~3) in Figure 1D.
Reviewer 3 Report
General comment
In this manuscript, authors reported the effectiveness of the RIG-G gene as a marker in the treatment of APL. The topics in the manuscript are appropriate and the findings in the manuscript will be of interest. However, there are many issues need to be clarified.
Major comment
#1 Quantitative real-time PCR analysis of PML-RARA mRNA using bone marrow specimens is already the mainstream in the treatment of APL, and Improvements of prognosis can be achieved by detecting molecular relapse at an early clinical stage and immediately treating with arsenous acid and so on. The authors should show a side-by-side comparison of quantitative real-time PCR analysis of PML-RARA mRNA and RIG-G gene expression in peripheral blood during the course of treatment of APL patients. Only by showing this result can it be said that RIG-G gene expression is a useful biomarker.
#2 All of the samples used in this analysis are from authors` affiliation. However, differences in RIG-G gene expression between ethnic and genetic polymorphism factors should be considered.
#3 The authors showed that treatment of APL cell lines with ATRA induces RIG-G gene expression, but I think it is necessary to show whether treatment of patient APL cells with ATRA actually increases RIG-G gene expression in vitro. It is also necessary to show in vitro data how RIG-G gene expression on normal monocytes fluctuates when treated with ATRA.
#4 In Figure 4A, the expression of the RIG-G gene in the peripheral blood of APL patients after ATRA treatment is increased, but which cells in APL patients are supposed to produce this gene? Isn`t it possible that RIG-G gene is produced by other than blood cells? Is there any difference between cases with high RIG-G expression and cases with relatively low RIG-G expression? The author needs to clarify these points.
Author Response
#1 Quantitative real-time PCR analysis of PML-RARα mRNA using bone marrow specimens is already the mainstream in the treatment of APL, and Improvements of prognosis can be achieved by detecting molecular relapse at an early clinical stage and immediately treating with arsenous acid and so on. The authors should show a side-by-side comparison of quantitative real-time PCR analysis of PML-RARα mRNA and RIG-G gene expression in peripheral blood during the course of treatment of APL patients. Only by showing this result can it be said that RIG-G gene expression is a useful biomarker.
--Thanks to the reviewer’s suggestion. Yes, showing a side-by-side comparison of quantitative real-time PCR analysis of PML-RARα mRNA and RIG-G gene expression in peripheral blood during the course of treatment of APL patients can it be said that RIG-G gene expression is a useful biomarker. It is well know that PML-RARα mutations occur in more than 95% of APL patients and disappear after treatment. In this study, we selected 20 APL patients at the initial onset of the disease, the bone marrow test results of PML-RARα gene were all positive. After a period of treatment, the bone marrow test PML-RARα turned negative (except the patients who died). However, it should be pointed out that there are many subtypes of the PML-RARα fusion gene, and the remission of molecular biology after induction and consolidation therapy is also inconsistent with the remission of cell morphology in the bone marrow. Therefore, this study temporarily selected the morphological changes of bone marrow cells in patients with APL as the reference index,and we have not paid continuous attention to the changes of PML-RARα mRNA gene in APL treatment. Of course, when we continue to collect new APL patients in the future, we will consider monitoring these two indicators at the same time to clarify the connection between the two, and establish the basis for the clinical application of RIG-G gene. This present study only indicated the expression of RIG-G gene in peripheral blood decreased in APL patients and increased after treatment with ATRA, RIG-G gene was found lower expression in peripheral blood and after the treatment of APL, and the detection of RIG-G gene expression in peripheral blood can effectively monitor the disease changes of APL patients and avoid harmful bone marrow puncture injury.
#2 All of the samples used in this analysis are from authors` affiliation. However, differences in RIG-G gene expression between ethnic and genetic polymorphism factors should be considered.
--Thanks to the reviewer’s suggestion. We have also considered the impact of this aspect. As one of the regions with the highest level of medical care in China, Shanghai itself attracts APL patients from all over the country. Although, we will continue to collect specimens from our partners (Northern China, Liaoning, Dalian) in follow-up studies, in order to use large samples to verify the feasibility of RIG-G as a marker for clinical application.
#3 The authors showed that treatment of APL cell lines with ATRA induces RIG-G gene expression, but I think it is necessary to show whether treatment of patient APL cells with ATRA actually increases RIG-G gene expression in vitro. It is also necessary to show in vitro data how RIG-G gene expression on normal monocytes fluctuates when treated with ATRA.
--Thanks to the reviewer’s suggestion. In the preliminary study of our cooperative laboratory, it has been proved that the changes in RIG-G gene expression after ATRA treatment of patients’ APL primary cells (Lou YJ, Pan XR, Jia PM, Jin J, Tong JH. RIG-G inhibits the proliferation of NB4 cells and propels ATRA-induced differentiation of APL cells. Leuk Res. 2016; 40:83-89. doi:10.1016/j.leukres.2015.11.007), the expression of RIG-G mRNA levels did increase with ATRA treatment.
#4 In Figure 4A, the expression of the RIG-G gene in the peripheral blood of APL patients after ATRA treatment is increased, but which cells in APL patients are supposed to produce this gene? Isn`t it possible that RIG-G gene is produced by other than blood cells? Is there any difference between cases with high RIG-G expression and cases with relatively low RIG-G expression? The author needs to clarify these points.
--Thanks to the reviewer’s suggestion. Our previous research results proved that the RIG-G gene is produced by re-differentiated promyelocytic cells after ATRA treatment (Xiao S, Li D, Zhu HQ, et al. RIG-G as a key mediator of the antiproliferative activity of interferon-related pathways through enhancing p21 and p27 proteins. Proc Natl Acad Sci U S A. 2006;103(44):16448-16453. doi:10.1073/pnas.0607830103). Since the number of samples observed in this study is only 20 cases, we have observed that APL patients with lower RIG-G expression have more blast cells, and treatment may take longer time, but this phenomenon is temporarily no statistically significant. The specific difference may require us to continue to collect APL patients and further verify with large samples in future.
Round 2
Reviewer 2 Report
Although the authors adequately responded to the answer posed in the original, their answers further weakened the potential applicability of this report to the field. Sequencing of a PCR product from a plasmid is not sufficient to conclude that the PCR fragment amplified from whole-genome samples is the predicted target gene. Consequently, they have not adequately established that their assay is monitoring RIG-G expression in patient samples.
The potential utility of this study is further limited by the authors' acknowledgment that there is only a trend, but not statistically significant correlation between RIG-G expression and changes in blast percentage. This is not unexpected for a small study, but this limitation is not addressed in the text. Rather, the authors state that "RIG-G gene expression in peripheral blood can effectively monitor disease changes in APL patients" which is contrary to the authors' response to reviewer comments.
Reviewer 3 Report
Basically, it's hard to say that they answered my comments side by side, and it's hard to say that the content has been brushed up.